# Optimal exit choice during highway tunnel evacuations based on the fire locations

**Yuchen Wang** [iD]**, Jianxiao Ma***, **Yuhang Liu, Yingjia Bai, Le Xu**

College of Automobile and Traffic Engineering, Nanjing Forestry University, Nanjing, China

* majx@njfu.edu.cn

**Data Availability Statement:** All relevant data are within the manuscript and its Supporting information files.

**Funding:** This research was funded by the Postgraduate Research & Practice Innovation

## Abstract

In the case of a fire, the choice of exit in the highway tunnel is strictly limited by fire location, which seriously affects the evacuation time. A spontaneous or disorderly exit choice might result in a decreased evacuation efficiency and utilization rate of exits. In this paper, we propose a strategy to obtain the optimal exit choice based on fire location during highway tunnel evacuations. In our strategy, first, the vehicle distributions and locations of evacuating occupants are determined in the traffic simulation program VISSIM. The evacuation simulation software BuildingEXODUS is employed to obtain the corresponding parameters of the evacuation process and analyze the impacts of different fire locations on the evacuation time. During the analysis, the optimal productivity statistics ($OPS$) is selected as the evaluation index. Then, the feature points of the crowding occupants are captured by the fuzzy c-means (FCM) cluster algorithm. Next, based on the feature points, the relationship between the location of the fire and boundary of the optimal exit choice under the optimal $OPS$ is obtained through the polynomial regression model. It is found that the R-squared($R^2$) and sum of squares for error (SSE) of the polynomial regression model, reflecting the accuracy estimation, are 98.02% and $2.79×10^{-4}$, respectively. Moreover, different fire locations impact the evacuation time of tunnel entrance and evacuation passageway. This paper shows that the location of the fire and boundary of optimal exit choice have a negative linear correlation. Taking the fire 110 m away from the evacuation passageway as an example, the $OPS$ of our strategy can be decreased by 35.6% when compared with no strategies. Our proposed strategy could be applied to determine the location of variable evacuation signs to help evacuating occupants make optimal exit choices.

## Introduction

With the rapid development of highways, traffic safety has become a key concern of the participants and managers [1, 2]. As a bottleneck section of highway, highway tunnels have a higher risk of being the site of crashes than open roads [3]. According to data from Jiangsu Expressway Co. Ltd., China, for a section of provincial highway S38 from Tianwang to Maoshan, the tunnel section accounts for approximately 2.25% of the whole line, while the accident rate was over 10% between 2015 and 2019. Once there is an accident caused by the spontaneous

Program of Jiangsu Province, code: KYCX20_0886.

**Competing interests:** The authors have declared that no competing interests exist.

combustion of a heavy truck, in the case of fire, it is easy to spread the fire fast and difficult to escape in the highway tunnel because of closed interior space, limited ventilation and adverse spatial structure [4, 5]. Thus, efficient use of exits during highway tunnel evacuations under the fire environment is an important research topic.

Although lots of attention is paid to the study of exit choice in the building and room [6, 7], there are few systematic studies of exit choice in the highway tunnel. On the one hand, the existing modelling methods of exit choice under the fire environment is concentrated on influencing factors through experiments, without considering the fire locations. For example, Fridolf et al. [8] studied the influence of different way-finding installations on exit choice in smoke-filled tunnels. Furthermore, Ronchi et al. [9] specially conducted the impact of the emergency exit portal complemented with information signs on exit choice. Lovreglio et al. [10] investigated the effect of environmental conditions (presence of smoke, emergency lighting and distance of exit) and social factors (interaction with evacuees) to build a mixed logit model for predicting exit choice. Meanwhile, they applied random utility theory to study the effect of herding behavior on exit choice [11]. Haghani et al. [12] studied spatial distance, congestion level, exit visibility and the size of moving flows on exit choice using under face-to-face interviews and field-type experiments. After that, they studied exit choice of crowds under high and low levels of urgency[13]. However, they mainly concentrated on buildings with a balanced ratio of length to width. And the fire is easier to spread in the highway tunnel. Therefore, more concerns and researches are needed to focus on the exit choice during highway tunnel under fire environment.

Due to the complexity of conducting field research, another method used to model exit choice is simulation modeling. Different simulation models have been established and used by researchers, such as BuildingEXODUS [14] and Pathfinder [15]. Cuesta et al. [16]observed children from 6 to 16 years of age during the evacuation exercises. They used the BuildingEXODUS to simulate the evacuation of children in school and verify the accuracy of the model predictions. Hunt et al. [17]used the BuildingEXODUS evacuation model to represent moving objects and simulated the evacuation of the patients and the staff in hospital. In addition, cellular automata [18, 19], ant colony algorithm [20] and agent-based model [21] are utilized to study influences of different factors on exit choice. These models are used to simulate the spontaneous behavior of human when making exit choice. Moreover, these approaches simulate the real spontaneous exit choice from the theoretical point of view to ensure the accuracy of the simulation.

However, the spontaneous exit choices of evacuating occupants sometimes cause the congested exits because of herding behavior [22]. Therefore, from the perspective of management, manual intervention on exit choice is of great application value to engineering compared with spontaneous exit choice. For example, broadcast and evacuation signs can be used to guide evacuating occupants at different locations to different evacuation exits, which can improve the evacuation efficiency and survival rate of evacuating occupants.

The aim in the study is to determine the optimal exit choice for different fire locations and evacuating occupants at different positions. We propose a strategy to obtain the optimal exit choice during highway tunnel evacuations based on fire location. The research framework includes three parts. First, we use BuildingEXODUS and VISSIM to obtain the parameters about vehicle distributions, locations of evacuating occupants, evacuation process and analyze the impacts of different fire locations on evacuation time. Then, the *OPS* output from BuildingEXODUS is selected as the evaluation index. Next, FCM cluster algorithm is used to capture the feature points of the occupants. Based on the feature points, the relationship between the location of fire and boundary of optimal exit choice under the optimal *OPS* is obtained through polynomial regression model.

The main contributions of this paper can be summarized as follows. First, we connect the vehicle distribution of VISSIM with the enclosure and movement modules of BuildingEXO-DUS to realize the exchanges of simulation data, which include output parameters about vehicle distributions, locations of evacuating occupants and evacuation process. Second, we make use of FCM cluster algorithm to capture the feature points of the occupants, which is valid to describe the characteristics of the crowding occupants, and the feature points could be convenient to calculate. Third, we apply the polynomial regression model to solve the optimal exit choice, which could provide an explicit strategy for evacuating occupants to determine optimal evacuation exit.

The remainder of this paper is organized as follows. Section 2 introduces problem statements. In Section 3, we explain the proposed method in detail. The input parameters and results of model case study in Maoshan Tunnel are analyzed in Section 4. Finally, conclusions are given in Section 5.

## Problem statement

In Fig 1, we demonstrate occupants evacuating in the highway tunnel. When a fire breaks out between the tunnel exit and evacuation passageway, evacuating occupants located upstream choose to drive away. The occupants located from fire to evacuation passageway are supposed to escape through the evacuation passageway while the occupants located from evacuation passageway to entrance may choose to escape through the evacuation passageway or entrance.

In the case of fire, the evacuating occupants are stimulated by the crisis environment and subconsciously flee to the entrance in the reverse direction, resulting in the inefficient utilization of evacuation passageway and longer evacuation time [23]. Therefore, it is necessary to study optimal exit choice for occupants during highway tunnel evacuations.

This work focuses on evacuating occupants in different positions to determine optimal evacuation exit. The *OPS* is used to measure the utilization rate of evacuation passageway and exit [24], and can be expressed as follows:

$$OPS = \frac{\sum_{e=1}^{E} (TET - EET_e)}{(E - 1) \cdot TET} \tag{1}$$

where *E* denotes the number of used exits, $EET_e$ is the time of the last occupant evacuating from exit e, and *TET* represents the total evacuation time, which is equal to the maximum value of $EET_e$.

The calculation of *OPS* includes not only total evacuation time, but also the time of the last occupant evacuating from exit, which reflects the utilization rate and evacuation efficiency. The range of *OPS* is from 0 to 1. In general, the lower *OPS* is, the more balanced and higher efficiency of evacuation is. If evacuating occupants ignore one of the exits or gather at the same evacuation exit, the index could reflect the low efficiency of evacuation.

Based on the above analysis, the effects of different fire locations and exit choices of evacuating occupants on *OPS* can be studied. The relationship between the location of the fire and boundary of optimal exit choice can be built by minimizing *OPS*. Furthermore, the boundary of optimal exit choice for occupants is determined, i.e., occupants located on the left of boundary evacuate through the evacuation passageway, while occupants located on the right should evacuate through the tunnel exit.

In Fig 2, we establish the mathematical model of occupants evacuating in the highway tunnel. As illustrated in Fig 2, the midpoint of the evacuation passageway is defined as the origin. $L_{EX}$ denotes the distance from origin to tunnel entrance, and $L_{ET}$ denotes the distance from

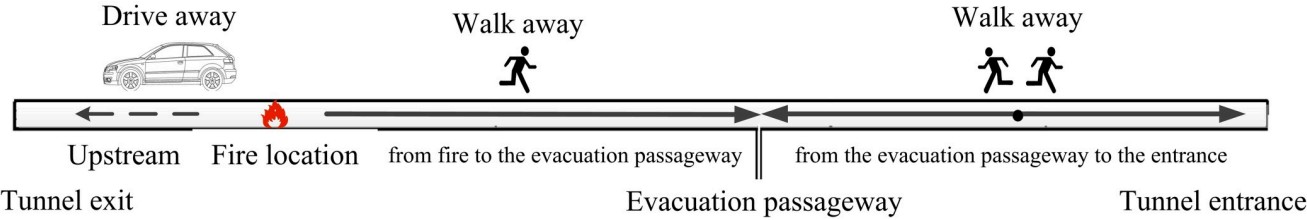

**Fig 1. Occupants evacuating in the highway tunnel.**

origin to fire location, respectively. $L_x$ means the distance from origin to boundary and $L_y$ represents the distance from origin to fire location, respectively. To avoid using the specific tunnel length as a constraint interval and increase the applicability of the model, we normalize the data $L_x$ and $L_y$ into the same interval, i.e., $x = L_x/L_{EX}$ and $y = L_y/L_{ET}$. $x$ describes the ratio of the distance from origin to boundary and that from origin to tunnel entrance, indicating the proportion of occupants evacuating from the evacuation passageway. The larger $x$ is, the more occupants evacuate from the evacuation passageway. $y$ represents the ratio of the distance from origin to fire location and that from origin to tunnel exit, describing how far the fire to the evacuation passageway. The larger $y$ is, the farther the fire is from the evacuation passageway.

## Method

### Simulation softwares

Due to safety concerns and the complexity of conducting field research, BuildingEXODUS [14] is employed in this study to simulate the evacuation process and evaluate the evacuation efficiency in the tunnel. BuildingEXODUS could repeatedly simulate the evacuation and solve the conflicts between field evacuation and traffic disruption, thereby saving costs and time [25]. It is adopted in modeling and can include occupants, actions, behaviors, disasters, and so forth [26].

The car-following model of VISSIM could easily simulate and extract the trajectories of vehicles[27, 28]. Therefore, the VISSIM is used to display vehicle distributions and operations in real time when the fire occurs [29], determining the locations of evacuating occupants and obstacles in the tunnel. The initial locations of evacuating occupants are calculated based on vehicle distributions. In addition, the initial locations of evacuating occupants and obstacles are associated with the enclosure and movement modules in BuildingEXODUS.

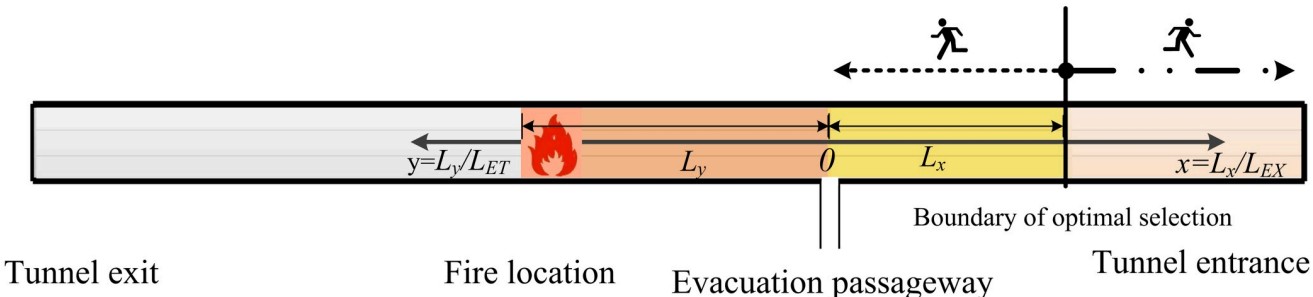

**Fig 2. Mathematical model of occupants evacuating in the highway tunnel.**

Note that the reliability of the evaluation results partly depends on the accuracy of model input, such as geometric parameters and traffic volumes. Under this condition, the input of model in this study should be carefully calibrated and validated based on monitoring data and literature data.

## FCM cluster algorithm

In the process of evacuation, the evacuating occupants tend to choose the most congested exit rather than another exit because they trust the majority of other occupants' behaviors and consider the congested exit to be the right exit [30]. Thus, the clustering phenomenon easily occurs. To capture the characteristics of the crowding occupants, the FCM algorithm employs the fuzzy theory in the determination of the membership relationship between data elements in the cluster algorithm[31]. It converts the hard division of the data elements in the traditional cluster algorithm with the membership degree between different clusters into soft division[32, 33].

The algorithm is described as follows. The sample space $P$ has $N$ data elements. The number of clusters is $C$. The objective function of the FCM algorithm $J_m$ is defined as

$$J_m = \sum_{i=1}^{N} \sum_{j=1}^{C} u_{ij}^m \|P_i - V_j\|^2, 1 < m < \infty \tag{2}$$

where $m$ is a fuzziness index, controlling the degree of blurring of the clustering results, $u_{ij}$ represents the membership of cluster $j$ of sample $i$, $P_i$ is the $i_{th}$ sample and $V_j$ denotes the centroid of cluster $j$, respectively.

The FCM algorithm minimizes the objective function $J_m$ by iteration. For a sample $P_i$, the sum of the membership degrees for each cluster is 1, that is,

$$\sum_{j=1}^{C} u_{ij} = 1, i = 1, 2, \cdots, N. \tag{3}$$

To find the minimum value of the Eq (2), the necessary conditions are as follows:

$$u_{ij} = \frac{\left(\frac{1}{\|P_i - V_j\|}\right)^{\frac{1}{m-1}}}{\sum_{j=1}^{C} \left(\frac{1}{\|P_i - V_j\|}\right)^{\frac{1}{m-1}}} \tag{4}$$

The termination condition of the iteration is shown below:

$$\max_{ij}\left\{u_{ij}^{(C+1)} - u_{ij}^{(C)}\right\} < \delta \tag{5}$$

where $\delta$ is the error threshold.

The clustering effect is evaluated using the Xie-Beni effectiveness index $V_{xie}$ [34]. The smaller the index is, the better the clustering effect is. The index $V_{xie}$ can be expressed as follows:

$$V_{xie} = \frac{\frac{1}{N}\sum_{i=1}^{C}\sum_{j=1}^{N} u_{ij}^m \|V_j - P_i\|^2}{\min_{i \neq j} \|V_j - V_i\|^2} \tag{6}$$

The FCM cluster method is used to classify different $x$ values based on $OPS$ and $y$. Then, the regression analysis is carried out for these feature points to research the relationship between $OPS$ and $x$, $y$.

## Polynomial regression model

Polynomial regression is a kind of linear regression [35, 36]. It is assumed that given $N$ data points, we search an appropriate polynomial of $q$-order to describe the relationship between independent variable $OPS$ and the dependent variable $x$, $y$. Since $x$ and $y$ affect $OPS$ simultaneously, the interaction between $x$ and $y$ is considered to have an impact on $OPS$. The general form of polynomial regression can be expressed as follows:

$$OPS = \beta_0 + \beta_1 y + \beta_2 x + \beta_3 x^2 + \beta_4 y^2 + \beta_5 xy \ldots + \beta_n y^q + \varepsilon \tag{7}$$

where $\varepsilon$ is the random deviations or residuals, $x$ and $y$ are the independent variables, $OPS$ refers to a dependent variable, $\beta_i$ is the coefficient of the polynomial and $q$ is the order of a polynomial, respectively.

Polynomial fitting is used to construct a $q$-order polynomial, and does not require all data points to be strictly. However, it is expected to pass most of these data points as many as possible, so that the residual error between the estimated parameter value and the actual value can be minimized.

In this paper, the curve fitting tool in MATLAB is exploited to estimate the parameters $\beta_i$. The following step is used to solve the relationship between $x$ and $y$ when $OPS = f(x,y)$ obtains the optimal value in region D ($0 < y \leq 1$, $0 \leq x \leq 1$).

Considering the above analysis, the optimal exit choice during highway tunnel evacuations can be expressed as follows:

$$Z(i) = \begin{cases} 0, i \in [0, x \cdot L_{EX}] \\ 1, i \in (x \cdot L_{EX}, L_{EX}] \end{cases}, x = f(y), \tag{8}$$

where $i$ is the location of the evacuating occupants; $Z(i)$ denotes that occupant located $i$ escapes from which evacuation passageway; $Z(i) = 0$ if evacuating occupants escape from evacuation passageway and 1 if evacuating occupants escape from the tunnel entrance; $L_{EX}$ refers to the distance from origin to tunnel entrance; $L_{ET}$ is the distance from origin to tunnel exit; $x$ stands for the ratio of the distance from origin to boundary and that from boundary to tunnel entrance; $y$ is the ratio of the distance from origin to fire location and that from origin to tunnel exit.

## Model case study

### The highway tunnel

Maoshan Tunnel is one of the earlier highway tunnels built in Jiangsu Province, China. It has been open to traffic for more than 10 years. Because the tunnel is short, it is designed without a ventilation system, resulting in non-flowing internal air. In the case of a fire, the dispersion of smoke tends to reduce the visibility of evacuating occupants and their perception of distance, increasing the difficulty of escaping. Thus, it is of great significance to select a reasonable and effective evacuation exit to improve the efficiency of highway tunnel.

The highway tunnel considered in this study is the westbound tunnel of Maoshan Tunnel, as illustrated in Fig 3. The total length of the tunnel is 585 m. There is one evacuation passageway between the exit and entrance, which is located 235 m from the entrance and 350 m from the exit. The width of the crossing is 1 m, and its height is 2.87 m.

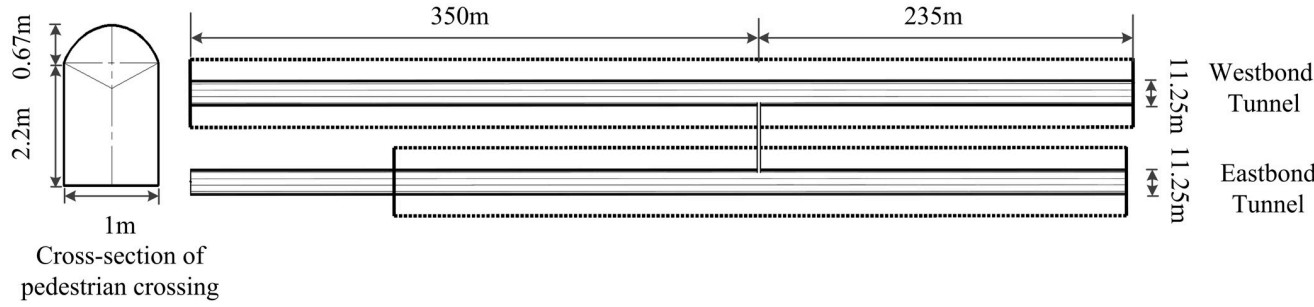

**Fig 3. Horizontal and cross section of Maoshan Tunnel.**

## Model input calibration

A set of simulation parameters is required to ensure the facticity of evacuation under the fire environment in the tunnel. In this study, monitoring data and literature data are collected to calibrate those simulation parameters input into BuildingEXODUS and VISSIM before simulation. Literature data are used to provide fire parameters and evacuation speed to ensure the authenticity of simulation. A previous emergency exercise of Maoshan Tunnel is used to validate the evacuation process in the highway tunnel under fire environment.

**Parameters of the fire.** The fire is an important input factor that should be carefully calibrated. The parameters of fire determined based on both monitoring and literature data are input into the hazard module of BuildingEXODUS. Assuming that the fire in this study is caused by gasoline combustion. The size of the fire is $4.6 \times 1.7 \times 1.5$ m³. The heat release rate is 50 MW and fire growth coefficient is 0.188 [37].

**Traffic volumes and vehicle distributions.** To study the evacuation under the worst circumstances, the maximum traffic volumes should be taken as the total number of vehicles. The traffic volumes and vehicle types of Maoshan Tunnel at three different time periods, morning peak, flat peak and evening peak were recorded for October 2019 (31 days), as is shown in Table 1. It can be seen that the maximum traffic volume was 1356 veh/h which appeared during the evening peak.

Vehicle distribution is used to describe the spatial location of vehicle in the tunnel, which is another key parameter to determine the initial locations of evacuating occupants. According to the previous emergency exercise of Maoshan Tunnel, the fire alarm rings 30 seconds after a fire occurs. It takes evacuating occupants 30 seconds to escape from the cars after noticing the fire. The maximum traffic volume is input into the VISSIM software to simulate the distribution of vehicles from freely flowing to congested traffic after the fire happens.

**Table 1. Traffic volumes and vehicle proportion at three different time periods.**

| Time periods | Parameter | Passenger vehicle | Coach | Heavy truck | Articulated vehicle | Traffic volumes |
|---|---|---|---|---|---|---|
| Flat peak | Traffic volumes | 571 | 27 | 151 | 32 | 781 |
| | vehicle proportion (%) | 73.1 | 3.5 | 19.3 | 4.1 | 100 |
| Morning peak | Traffic volumes | 1000 | 53 | 222 | 28 | 1303 |
| | Vehicle proportion (%) | 76.7 | 4.1 | 17.1 | 2.1 | 100 |
| Evening peak | Traffic volumes | 1131 | 26 | 160 | 39 | 1356 |
| | Vehicle proportion (%) | 83.4 | 1.6 | 10.3 | 2.8 | 100 |

**The number and initial locations of evacuating occupants.** The number of evacuating occupants is determined by the maximum traffic volumes, vehicle proportion, the capacity of different vehicles and passenger proportion of vehicles, and can be expressed as follows:

$$Q = \sum_{i=1}^{4} \left( \mu_i \times m_i \times \eta_i \times n \right) \tag{9}$$

where $Q$ is the number of evacuating occupants; $\mu_1$, $\mu_2$, $\mu_3$, and $\mu_4$ denote the vehicle proportion of passenger vehicles, coaches, heavy trucks and articulated vehicles, respectively; $m_1$, $m_2$, $m_3$, and $m_4$ are the capacity of passenger vehicles, coaches, heavy trucks and articulated vehicles; $\eta_1$, $\eta_2$, $\eta_3$, and $\eta_4$ refer to passenger proportion of passenger vehicles, coaches, heavy trucks and articulated vehicles; $n$ represents maximum traffic volumes.

These parameters in (9) are determined by monitoring data. There are four types of vehicles: articulated vehicles, heavy trucks, coaches and passenger vehicles. The outer dimensions and capacity of different vehicle types are determined by the Chinese Technical Standard of Highway Engineering, as shown in Table 2.

The passenger proportion of vehicles is set to 50% for passenger vehicles, 90% for coaches, and 100% for heavy trucks and articulated vehicles. Meanwhile, results of the calculation are rounded to integers. The initial locations of evacuating occupants are calculated by vehicle distributions.

**Evacuation speed.** The evacuation speed in a dark tunnel is extremely slow and must be determined accurately to assess tunnel fire safety. According to [38], age and gender have little effect on evacuation speed. Evacuation speeds of evacuating occupants are approximately log-normal within a 95% interval (using the 2.5th and 97.5th percentiles of the distribution as endpoints, which are 0.24 and 0.88 m/s with a mean value of 0.49 m/s). In this study, the average speed is 0.49 m/s, which is input into the movement module of BuildingEXODUS to simulate the movement of occupants in the highway tunnel under fire environment.

The analysis is conducted based on the above parameter settings. When a fire breaks out in the evacuation passageway, evacuating occupants cannot evacuate through it. Due to the size of fire, 2 m away from the evacuation passageway ($y = 0.05$) is selected as the location of the initial fire. To appropriately indicate the trend of $OPS$, $y$ varies with a step of 0.1. Meanwhile, as the crowd is concentrated, for different $y$, $x$ varies with a step of 0.05. Finally, the corresponding $x$, $y$ and $OPS$ are outputted from BuildingEXODUS (S1 Table). The process of solving $x$, $y$ and the corresponding $OPS$ is shown in Fig 4.

## Results and discussion

### The impact of the fire location

The location of fire has a significant impact on the evacuation time [39]. The paper delves into the impact of the distance between the fire and evacuation passageway on the evacuation time of tunnel entrance and evacuation passageway. The results are shown in Fig 5.

As shown in Fig 5, when $y$ varies from 0.05 to 0.35, the evacuation time for the evacuation passageway and tunnel entrance changes significantly with $x$. In contrast, as $y$ varies from 0.45 to 1, there is less significant change with $x$ in the evacuation time for the evacuation passageway and tunnel entrance. This indicates that the impacts on the evacuation time in the evacuation passageway and tunnel entrance vary according to different fire locations. The shorter the distance between the location of fire and origin is, the greater the fluctuation of the evacuation time is. As the distance between the location of fire and origin gets farther, the total evacuation time also increases. The potential reason may be that more occupants trapped from fire to evacuation

**Table 2. The outline dimensions and capacity of different vehicle types.**

| Vehicle types | Length | Width | Height | Capacity |
|---|---|---|---|---|
| Passenger vehicle | 6 | 1.8 | 2 | 6 |
| Coach | 13.7 | 2.55 | 4 | 50 |
| Heavy truck | 12 | 2.5 | 4 | 2 |
| Articulated vehicle | 18.1 | 2.55 | 4 | 2 |

passageway may choose to escape through the pedestrian passageway. This phenomenon could decrease the capacity of the evacuation passageway and increase the total evacuation time.

## Optimal cluster number and coordinates

To find the most suitable cluster according to Eq (6), we calculate $V_{xie}$ of each cluster (S1 Code), as shown in Fig 6, where the number of clusters $C$ varies from 2 to 11.

It can be seen from Fig 6 that when clusters number $C$ is 7, $V_{xie}$ reaches the minimum value, corresponding to the optimal cluster number $C^*$, as shown in Fig 6. Therefore, the optimal cluster number is 7. The corresponding $x$, $y$, and $OPS$ coordinates are (0.13,0.25,0.55), (0.05,0.87,0.29), (0.25,0.32,0.16), (0.09,0.55,0.12), (0.40,0.10,0.36), (0.18,0.90,0.53) and (0.45,0.33,0.53), respectively.

## The relationship between $x$ and $y$ based on the optimal $OPS$

The relationship between $OPS$ and $x$, $y$ is established with the polynomial regression model. In Fig 7, we show the polynomial regression model of $x$, $y$ and $OPS$. There is a correlation between $x$ and $y$. The estimation results of each parameter are shown in Table 3.

From Table 3, the $R^2$ and SSE of the model are 98.02% and $2.79 \times 10^{-4}$, respectively. The dependent variable can be estimated from the model, which is generally available with SSE close to 0. Therefore, the regression model under the influence of $x$ and $y$ can be expressed as follows:

$$OPS = 2.45 - 9.91x - 5.52y + 12.62x^2 + 9.71xy + 3.52y^2 \tag{10}$$

The closer $OPS$ is to 0, the more balanced and higher efficiency of evacuation. The relationship between $x$ and $y$ is shown in Fig 8 by drawing the curves of different $y$ when $OPS$ approaches 0.

Since fire detectors in the highway tunnel can find the location of the fire [40], $y$ is known and we take the first and second derivatives of the model $OPS = f(x, y)$ with respect to $x$ in region D ($0 < y \leq 1, 0 \leq x \leq 1$), i.e.,

$$\frac{\partial OPS}{\partial x} = -9.91 + 25.24x + 9.71y,$$

$$\frac{\partial^2 OPS}{\partial x^2} = 25.24 > 0. \tag{11}$$

Then, letting $\frac{\partial OPS}{\partial x} = 0$ to solve the relationship between $x$ and $y$ under the condition of optimal $OPS$, we obtain

$$x = -\frac{9.91 - 9.71y}{2 \times 12.62} \approx -0.38y + 0.39, x \in [0.01, 0.39] \tag{12}$$

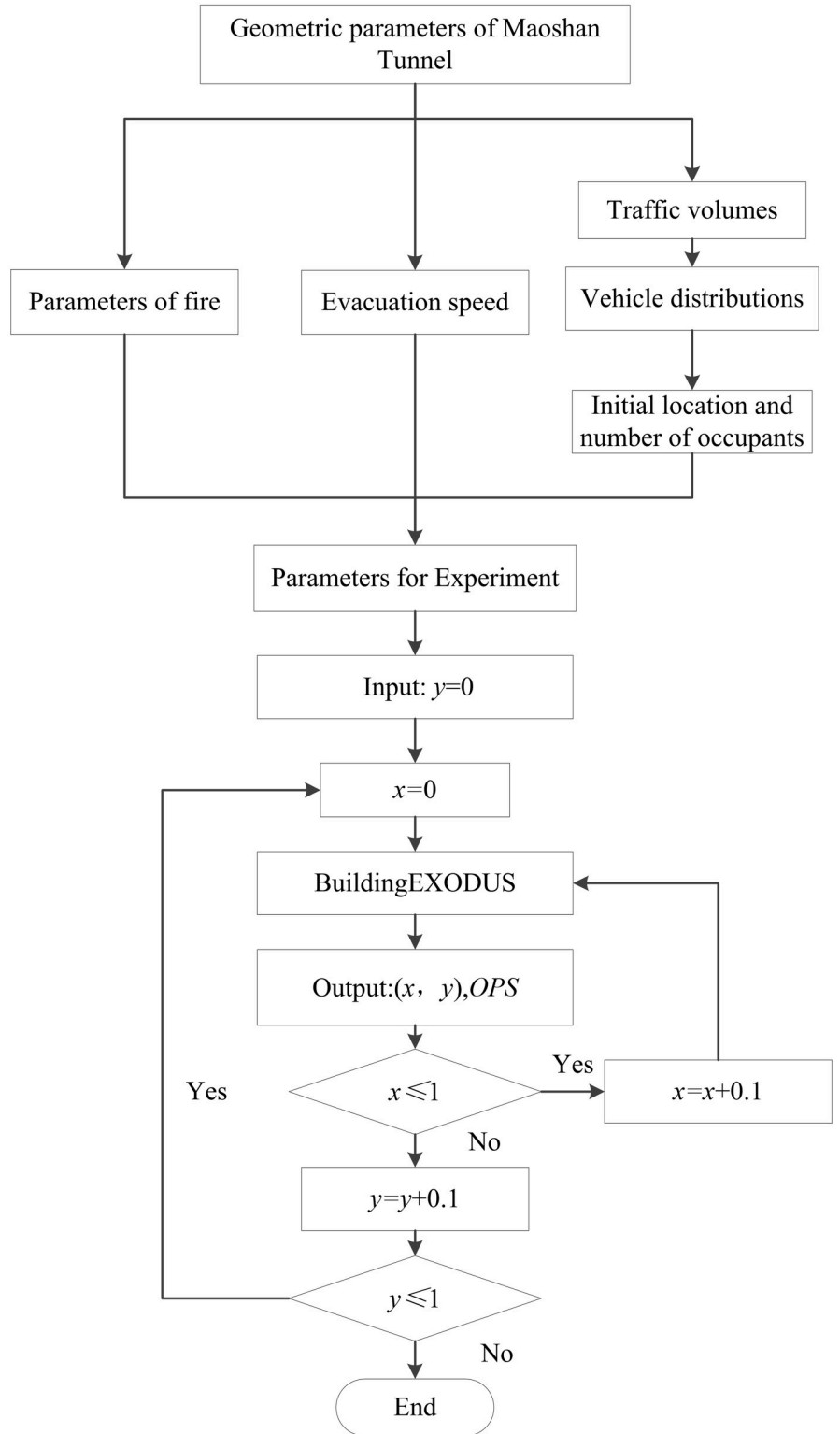

**Fig 4. Process of solving *x*, *y* and the corresponding *OPS*.**

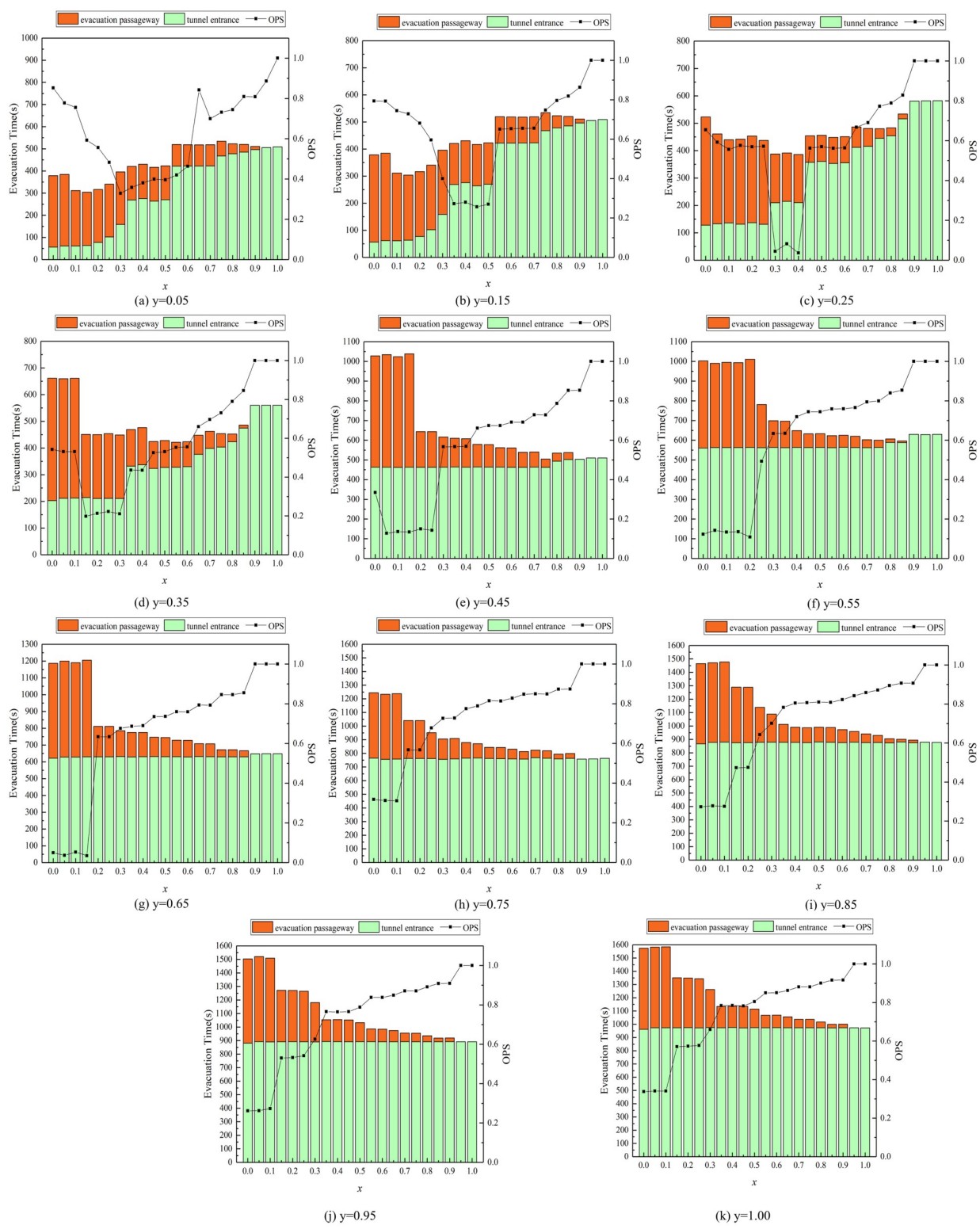

**Fig 5. Evacuation time and *OPS* of tunnel entrance and evacuation passageway.**

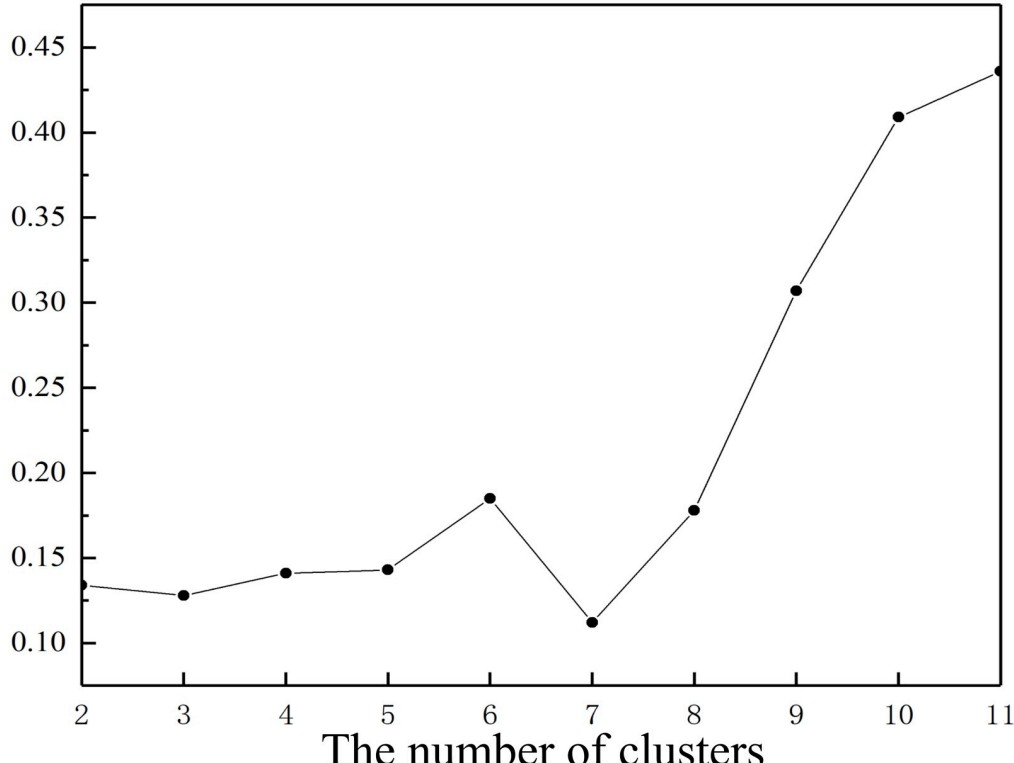

**Fig 6. $V_{xie}$ of different clusters.**

It can be seen in (12) that $y$ and $x$ have a negative linear correlation, which means that the farther the fire occurs away from the evacuation passageway, the closer the boundary is to the evacuation passageway, and the fewer occupants evacuate through the evacuation passageway. This is consistent with the previous study on fire location and evacuation time. Therefore, the

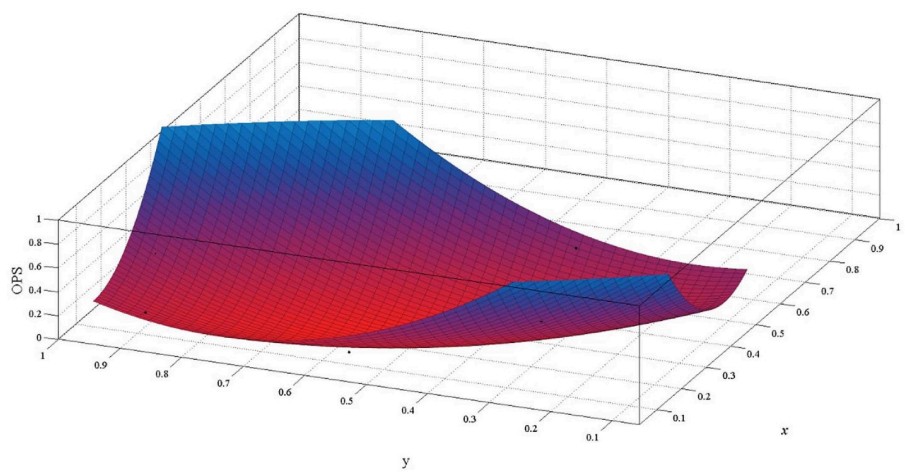

**Fig 7. Polynomial regression model of $x$, $y$ and $OPS$.**

**Table 3. The results of each parameter.**

| Parameter | Estimation Value | Confidence interval |
|:---:|:---:|:---:|
| $\beta_0$ | 2.45 | (2.28, 4.72) |
| $\beta_1$ | -9.91 | (-13.92, -3.37,) |
| $\beta_2$ | -5.52 | (-16.96, 5.91) |
| $\beta_3$ | 12.62 | (11.35, 13.60) |
| $\beta_4$ | 9.71 | (-10.36, 29.77) |
| $\beta_5$ | 3.52 | (1.07, 4.11) |

optimal exit choice during highway tunnel evacuations can be expressed as follows:

$$Z(i) = \begin{cases} 0, i \in [0, x \cdot L_{EX}] \\ 1, i \in (x \cdot L_{EX}, L_{EX}] \end{cases}, x = -0.38y + 0.39 \qquad (13)$$

If the fire occurs 110 m away from the evacuation passageway ($y = 0.31$), $x$ is calculated as 0.27 from Eq (13). Thus, occupants located 63 m away from the origin escape through the evacuation passageway and those who are 63–235 m away from the origin escape through the

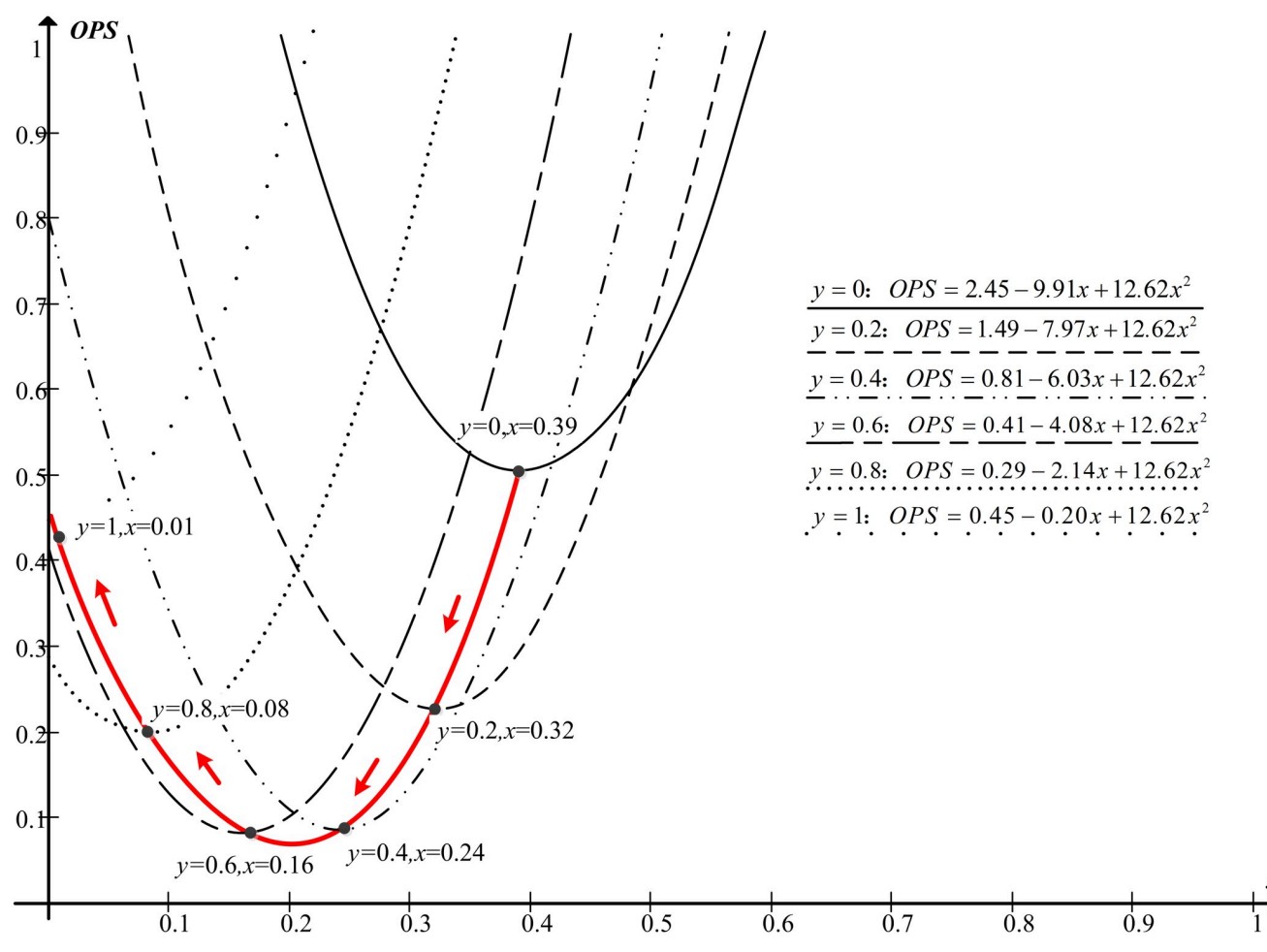

**Fig 8. Different y for $OPS = f(x, y)$.**

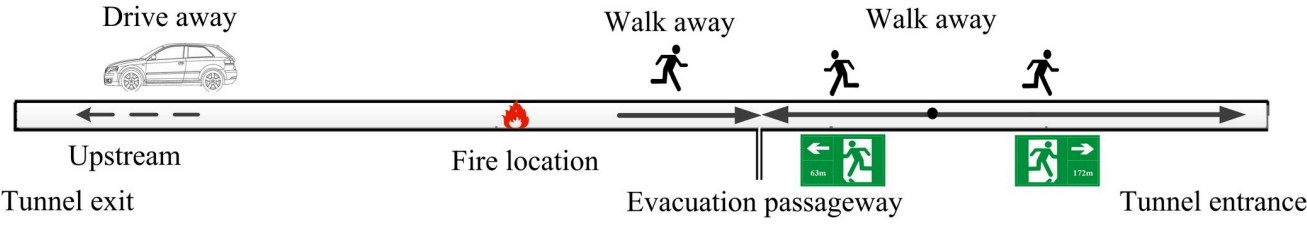

**Fig 9. Arrangement of evacuation signs.**

entrance of the tunnel. The *OPS* of our strategy is 0.195, while the *OPS* of a spontaneous and disorderly exit choice is 0.303, lowering *OPS* by 35.6%. Our strategy not only increases the utilization rate of exits, but also effectively improve the evacuation efficiency when compared with the spontaneous and disorderly exit choice.

According to the results of this study, the location of variable evacuation signs can be adopted to prompt, as shown in Fig 9.

## Conclusion

In this study, taking a highway tunnel in China as an example, we propose a strategy to choose the optimal exit during highway tunnel evacuations based on the fire location. First, we combine the vehicle distribution of VISSIM with the enclosure and movement modules of BuildingEXODUS to utilize the exchanges of simulation data. Next, the feature points of the crowding occupants are captured by the FCM cluster algorithm when the cluster number is 7. Finally, we apply the polynomial regression model to solve the optimal exit choice based on the feature points. It is found that the $R^2$ and SSE of the polynomial regression model, reflecting the accuracy estimation, are 98.02% and $2.79 \times 10^{-4}$, respectively.

Different fire locations are found to impact the evacuation time via evacuation passageway and entrance. The shorter the distance between the location of fire and the origin (the midpoint of the evacuation passageway) is, the greater the fluctuation in the evacuation time is. Under this circumstance, the location of the fire and boundary of optimal exit choice have a negative linear correlation, indicating that the farther the fire occurs from the evacuation passageway, the fewer occupants evacuate through the evacuation passageway. Taking a fire 110 m away from the evacuation passageway as an example, the *OPS* of our strategy can be decreased by 35.6% compared with no strategy. Overall, the location of the fire is detected by fire detectors in the highway tunnel. The optimal exit choice during highway tunnel evacuations based on fire location is obtained by our strategy, which could provide valuable information for optimal exit choice and arranging variable evacuation signs in the highway tunnel. In our study, vehicle distribution is an important factor that determines the spatial location of vehicle and initial locations of evacuating occupants. Hence, effective information warning and traffic guidance could reduce the number of trapped vehicles and reduce accident casualties. In the future, we could study the optimal exit choice based on the fire locations, considering several evacuation passageways between the exit and entrance. Alternatively, we could study the impact of different fire parameters on the optimal exit choice of the highway tunnel.

## Supporting information

**S1 Table. All data of *OPS*, *x* and *y*.**
(XLSX)

**S1 Code. Code for FCM of data.**
(PY)

## Acknowledgments

The authors would like to thank Jiangsu Expressway Co. Ltd. for providing the emergency exercise and related data for the case study.

## Author Contributions

**Conceptualization:** Yuchen Wang, Jianxiao Ma.

**Data curation:** Yuchen Wang, Yingjia Bai.

**Formal analysis:** Yuchen Wang, Yuhang Liu.

**Methodology:** Yuchen Wang, Jianxiao Ma.

**Software:** Yuchen Wang, Le Xu.

**Validation:** Yuhang Liu.

**Writing – original draft:** Yuchen Wang, Jianxiao Ma.

**Writing – review & editing:** Yuchen Wang, Le Xu.

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
