## [Decision Letter · Decision Letter 0]

13 Jul 2021

PONE-D-21-19452

Optimal Exit Choice During Highway Tunnel Evacuations Based on The Fire Locations

PLOS ONE

Dear Dr. ma,

Thank you for submitting your manuscript to PLOS ONE. After careful consideration, we feel that it has merit but does not fully meet PLOS ONE’s publication criteria as it currently stands. Therefore, we invite you to submit a revised version of the manuscript that addresses the points raised during the review process.

We look forward to receiving your revised manuscript.

Kind regards,

Feng Chen

Academic Editor

PLOS ONE

Journal Requirements:

Reviewers' comments:

Reviewer's Responses to Questions

**Comments to the Author**

1. Is the manuscript technically sound, and do the data support the conclusions?

Reviewer #1: Yes

Reviewer #2: Yes

2. Has the statistical analysis been performed appropriately and rigorously? 

Reviewer #1: Yes

Reviewer #2: Yes

3. Have the authors made all data underlying the findings in their manuscript fully available?

Reviewer #1: Yes

Reviewer #2: No

4. Is the manuscript presented in an intelligible fashion and written in standard English?

Reviewer #1: Yes

Reviewer #2: Yes

5. Review Comments to the Author

Reviewer #1: This paper proposes an exit selection strategy in highway tunnels under the a fire event. This research is of practical importance for safety management. The logic of this paper is clear, but there are still some issues needed to revise. The comments of the paper are as follows:

(1)The authors used VISSIM to simulate the vehicle distributions when the fire happened. Please explain the advantage of VISSIM.

(2)The research framework should be presented to indicate the methodology and contents.

(3) The rows in Table 2 should be spaced equally.

(4)The flowchart in Figure 4 of the paper is too cumbersome. The authors are advised to streamline it.

(5)The images inserted in the paper are blurry. It is recommended to use clear pictures.

(6)The format of the references should be consistent. For example, the name of a paper with only reference 1 is capitalized.

Reviewer #2: The study investigated an important topic using a simulation based approach. The authors combine two simulation tools for understanding the efficiency of highway tunnel evacuations and assisting the location of evacuation signs. Overall, I found the study interesting and would be of interest to broad audience. A few minor issues need to be corrected before it can be accepted for publication.

1. It is not clear how parameters in VISSIM and BuildingEXODUS are calinbrated, and how the simulation results are validated. The whole calibration process for simulation based study is necessary and the author may want to inlcude more details in supporting information.

2. Only the optimal productivity statistics is used as the index for evaluating the evacuation efficiency. I believe there are many other indexes or metrics that may work in a similar way. The authors should discuss further on the reasoning behind the choice of the index and potentially include the results from other metrics.

3. The figures submitted are of very poor quality. This should be significantly improved. Vectorized images are preferred for scientific publications.

4. The current discussion on major findings is pretty slim. The authors should include more detailed discussion on how the results can inform better evacuation prepration in highway tunnels and what are the major lessons learned from the case study that can be carried over other scenarios. The authors should also discuss the limitations, which is important for simulation based studies.

6. PLOS authors have the option to publish the peer review history of their article (what does this mean?). If published, this will include your full peer review and any attached files.

Reviewer #1: No

Reviewer #2: No

---

## [Author Response · Author response to Decision Letter 0]

6 Aug 2021

Dear reviewers,

Thanks very much for taking the time to review this manuscript. I really appreciate all your comments and suggestions! Please find my itemized responses below and my revisions in the re-submitted files.

Thank you very much for your consideration.

Best regards!

Jianxiao Ma

Reviewer #1:

1. The authors used VISSIM to simulate the vehicle distributions when the fire happened. Please explain the advantage of VISSIM.

Response: Thanks for your suggestions. We have added papers 27 and 28 with more detail about the advantage of VISSIM in the Method section. The specific revises are as follows (Word Lines: 153-156):

The car-following model of VISSIM could easily simulate and extract the trajectories of vehicles[27, 28]. Therefore, VISSIM is used to display vehicle distributions and operations in real time when the fire occurs [29], determining the locations of evacuating occupants and obstacles in the tunnel.

2. The research framework should be presented to indicate the methodology and contents.

Response: Thank you for your questions. We have added more descriptions to indicate the methodology and contents. The specific revises are as follows (Word Lines: 78-87):

The aim in the study is to determine the optimal exit choice for different fire locations and evacuating occupants at different positions. We propose a strategy to obtain the optimal exit choice during highway tunnel evacuations based on fire location. The research framework includes three parts. First, we use BuildingEXODUS and VISSIM to obtain the parameters about vehicle distributions, locations of evacuating occupants, evacuation process and analyze the impacts of different fire locations on evacuation time. Then, the OPS output from BuildingEXODUS is selected as the evaluation index. Next, FCM cluster algorithm is used to capture the feature points of the occupants. Based on the feature points, the relationship between the location of fire and boundary of optimal exit choice under the optimal OPS is obtained through polynomial regression model. 

3. The rows in Table 2 should be spaced equally.

Response: Thank you for your questions. In the revised version, we have modified Table 2(Word Lines: 278).

4. The flowchart in Figure 4 of the paper is too cumbersome. The authors are advised to streamline it.

Response: Thank you for your questions. In the revised version, we have modified Figure 4.

5. The images inserted in the paper are blurry. It is recommended to use clear pictures.

Response: Thank you for your questions. In the revised version, we have provided figures of high quality.

6. The format of the references should be consistent. For example, the name of a paper with only reference 1 is capitalized.

Response: Thank you for your questions. In the revised version, we have modified the format of the references.

Reviewer #2:

1. It is not clear how parameters in VISSIM and BuildingEXODUS are calinbrated, and how the simulation results are validated. The whole calibration process for simulation based study is necessary and the author may want to inlcude more details in supporting information.

Response: Thank you for your questions. In the revised version, we have added more descriptions to expand the calibration process. The specific revises are as follows:

A set of simulation parameters is required to ensure the facticity of evacuation under the fire environment in the tunnel. In this study, monitoring data and literature data are collected to calibrate those simulation parameters input into BuildingEXODUS and VISSIM before simulation. Literature data are used to provide fire parameters and evacuation speed to ensure the authenticity of simulation. A previous emergency exercise of Maoshan Tunnel is used to validate the evacuation process in the highway tunnel under fire environment. (Word Lines: 236-241)

The parameters of fire determined based on both monitoring and literature data are input into the hazard module of BuildingEXODUS. (Word Lines: 244-246)

The maximum traffic volume is input into the VISSIM software to simulate the distribution of vehicles from freely flowing to congested traffic after the fire happens. (Word Lines: 260-262)

In this study, the average speed is 0.49 m/s, which is input into the movement module of BuildingEXODUS to simulate the movement of occupants in the highway tunnel under fire environment. (Word Lines: 288-291)

2. Only the optimal productivity statistics is used as the index for evaluating the evacuation efficiency. I believe there are many other indexes or metrics that may work in a similar way. The authors should discuss further on the reasoning behind the choice of the index and potentially include the results from other metrics.

Response: Thank you for your questions. There are some evaluation indexes used to evaluate evacuation efficiency, such as evacuation time. The calculation of OPS includes not only total evacuation time, but also the time of the last occupant evacuate from exit, which reflects the utilization rate and evacuation efficiency. From the perspective of management, the OPS could be used to measure the utilization rate of evacuation passageway and exit. In the revised version, we have added more descriptions to explain the reason for using the index. The specific revises are as follows (Word Lines: 120-121):

The calculation of OPS includes not only total evacuation time, but also the time of the last occupant evacuating from exit, which reflects the utilization rate and evacuation efficiency.

3. The figures submitted are of very poor quality. This should be significantly improved. Vectorized images are preferred for scienific publications.

Response: Thank you for your questions. In the revised version, we have provided figures of high quality.

4. The current discussion on major findings is pretty slim. The authors should include more detailed discussion on how the results can inform better evacuation preparation in highway tunnels and what are the major lessons learned from the case study that can be carried over other scenarios. The authors should also discuss the limitations, which is important for simulation based studies.

Response: Thank you for your questions. In the revised version, we rewrote conclusion section. The specific revises are as follows (Word Lines: 383-394):

Overall, the location of the fire is detected by fire detectors in the highway tunnel. The optimal exit choice during highway tunnel evacuations based on fire location is obtained by our strategy, which could provide valuable information for optimal exit choice and arranging variable evacuation signs in the highway tunnel. In our study, vehicle distribution is an important factor that determines the spatial location of vehicle and initial locations of evacuating occupants. Hence, effective information warning and traffic guidance could reduce the number of trapped vehicles and reduce accident casualties. In the future, we could study the optimal exit choice based on the fire locations, considering several evacuation passageways between the exit and entrance. Alternatively, we could study the impact of different fire parameters on the optimal exit choice of the highway tunnel.

Editor:

1. We suggest you thoroughly copyedit your manuscript for language usage, spelling, and grammar. If you do not know anyone who can help you do this, you may wish to consider employing a professional scientific editing service.

Response: We have employed a professional scientific editing service American Journal Experts（AJE）to provide language editing.

2. Please include captions for your Supporting Information files at the end of your manuscript, and update any in-text citations to match accordingly. Please see our Supporting Information guidelines for more information:

Response: We have added captions of Supporting Information files at the end of your manuscript and updated any in-text citations to match accordingly.

Special thanks to you for your good comments. We have carefully revised and polished the manuscript, please the reviewer to check.

---

## [Decision Letter · Decision Letter 1]

9 Aug 2021

Optimal Exit Choice During Highway Tunnel Evacuations Based on The Fire Locations

PONE-D-21-19452R1

Dear Dr. ma,

We’re pleased to inform you that your manuscript has been judged scientifically suitable for publication and will be formally accepted for publication once it meets all outstanding technical requirements.

Kind regards,

Feng Chen

Academic Editor

PLOS ONE

Additional Editor Comments (optional):

Reviewers' comments:

Reviewer's Responses to Questions

**Comments to the Author**

1. If the authors have adequately addressed your comments raised in a previous round of review and you feel that this manuscript is now acceptable for publication, you may indicate that here to bypass the “Comments to the Author” section, enter your conflict of interest statement in the “Confidential to Editor” section, and submit your "Accept" recommendation.

Reviewer #1: All comments have been addressed

Reviewer #2: All comments have been addressed

2. Is the manuscript technically sound, and do the data support the conclusions?

Reviewer #1: Yes

Reviewer #2: Yes

3. Has the statistical analysis been performed appropriately and rigorously? 

Reviewer #1: Yes

Reviewer #2: Yes

4. Have the authors made all data underlying the findings in their manuscript fully available?

Reviewer #1: Yes

Reviewer #2: No

5. Is the manuscript presented in an intelligible fashion and written in standard English?

Reviewer #1: Yes

Reviewer #2: Yes

6. Review Comments to the Author

Reviewer #1: My comments have been addressed by authors. The revised version is suitable for the journal. So my suggestion is Accept.

Reviewer #2: (No Response)

7. PLOS authors have the option to publish the peer review history of their article (what does this mean?). If published, this will include your full peer review and any attached files.

Reviewer #1: No

Reviewer #2: No

---

## [Editor Report · Acceptance letter]

12 Aug 2021

PONE-D-21-19452R1 

Optimal Exit Choice During Highway Tunnel Evacuations Based on The Fire Locations 

Dear Dr. ma:

I'm pleased to inform you that your manuscript has been deemed suitable for publication in PLOS ONE. Congratulations! Your manuscript is now with our production department. 

Kind regards, 

on behalf of

Dr. Feng Chen 

Academic Editor

PLOS ONE